# Using Recurrent Neural Network to Optimize Electronic Nose System with Dimensionality Reduction

**Yanan Zou [1] and Jianhui Lv [2,\*]**

[1]  School of Science, Jilin Institute of Chemical Technology, Jilin 132022, China; zouyanan@jlict.edu.cn
[2]  International Graduate School at Shenzhen, Tsinghua University, Shenzhen 518055, China
\*  Correspondence: lvjianhui2012@sz.tsinghua.edu.cn

**Abstract:** Electronic nose is an electronic olfactory system that simulates the biological olfactory mechanism, which mainly includes gas sensor, data pre-processing, and pattern recognition. In recent years, the proposals of electronic nose have been widely developed, which proves that electronic nose is a considerably important tool. However, the most recent studies concentrate on the applications of electronic nose, which gradually neglects the inherent technique improvement of electronic nose. Although there are some proposals on the technique improvement, they usually pay attention to the modification of gas sensor module and barely consider the improvement of the last two modules. Therefore, this paper optimizes the electronic nose system from the perspective of data pre-processing and pattern recognition. Recurrent neural network (RNN) is used to do pattern recognition and guarantee accuracy rate and stability. Regarding the high-dimensional data pre-processing, the method of locally linear embedding (LLE) is used to do dimensionality reduction. The experiments are made based on the real sensor drift dataset, and the results show that the proposed optimization mechanism not only has higher accuracy rate and stability, but also has lower response time than the three baselines. In addition, regarding the usage of RNN model, the experimental results also show its efficiency in terms of recall ratio, precision ratio, and F1 value.

**Keywords:** electronic nose; recurrent neural network; dimensionality reduction; locally linear embedding

---

## 1. Introduction

Just as image processing originates from the sense of sight, electronic nose is inspired by the sense of smell. In fact, electronic nose (e.g., odor sensor, aroma sensor, mechanical nose, flavor sensor, multi-sensor array, artificial nose, odor sensing system, and electronic olfactometry) is an electronic olfactory system constructed to mimic the biological olfactory mechanism, which also belongs to the important scientific field of artificial intelligence (AI) [1,2]. The whole electronic nose system is usually composed of three modules: gas sensor, data pre-processing, and pattern recognition [3]. At present, the field of electronic nose has attracted worldwide attention, which proves that electronic nose has an important influence on the progress of human society [4]. However, currently, most studies concentrate on the applications of electronic nose such as quality inspection of agricultural and food products, dendrobium classification, classification and evaluation of quality grades of organic green teas, early detection of fish degradation, etc., irrespective of the inherent technique improvement of electronic nose. To the best of our knowledge, although there are some proposals to optimize the inherent technique, they usually focus on modifying the gas sensor module (doing measurements on cross-sensitivity of a variety of gases by sensor array) and barely pay attention to the improvement of data pre-processing and pattern recognition.

The research of electronic nose also belongs to AI field, which results from that data pre-processing and pattern recognition modules strongly rely on AI-related algorithms due to the natural ability

---

to process and comprehend large amounts of data, calibrate gas sensor array, and provide accurate classification and recognition results. In other words, data pre-processing and pattern recognition modules play indispensable roles in the electronic nose system. Without them, it would be very difficult or even impossible for electronic noses to have an intelligent effect. Furthermore, for AI, the process is usually divided into four stages: data collection, modeling, training, and evaluation [5]. Among them, the first two stages are of primary importance. However, the two stages to handle data pre-processing and pattern recognition in the electronic nose system face some limitations, such as low accuracy and stability, along with high response time. As a result, it is of great importance to further study data pre-processing and pattern recognition modules of electronic nose, depending on AI technique.

Regarding the data pre-processing module, its main responsibility is to do dimensionality reduction for high-dimensional data features extracted from gas sensor array, which facilitates computation and visualization by abandoning the redundant information. There are many methods for dimensionality reduction, and the typical representatives are principal component analysis (PCA) [6], linear discriminant analysis (LDA) [7], Laplacian eigenmaps (LE) [8], t-stochastic neighbor embedding (t-SNE) [9], and locally linear embedding (LLE) [10]. Among them, the first two methods belong to linear mapping, which cannot reach the efficient adaptivity as they need to manually adjust the threshold of cumulative interpretable variance. The last three methods belong to nonlinear mapping, which not only overcome the shortcoming of PCA and LDA, but also show more accurate dimensionality reduction to ensure data atomicity. However, LE and t-SNE have higher computation complexity and consume more computation time than LLE. In fact, the response time in the electronic nose system is an important metric, and the computation complexity has a great influence on response time. Thus, this paper uses LLE to reduce data dimensionality instead of PCA, LDA, LE, and t-SNE.

The pattern recognition module in the electronic nose system usually employs some AI algorithms to realize the related classification function. From the perspective of recognition form, there are four kinds of pattern recognition methods [11]: statistical pattern recognition [12], structural pattern recognition [13], fuzzy pattern recognition [14], and neural network (NN)-based pattern recognition [15]. With the rapid development of AI field, NN-based pattern recognition has overwhelming popularity and attracts much attention from the global researchers including those in the field of electronic nose. Furthermore, according to different structures, NN can usually be divided into artificial NN (ANN) [16], convolutional NN (CNN) [17], deep NN (DNN) [18], graph NN (GNN) [19], and recurrent NN (RNN) [20]. Among them, unlike ANN, CNN, DNN, and GNN, RNN includes the special circulation operation which can handle the input at the same time as storing information. In other words, RNN shows the obvious advantages in learning sequence and tree structure aspects, such as natural scene image and natural language processing. In fact, electronic nose is used to capture the natural environment state to do classification detection, which can be addressed by the technique of natural language processing. Given that, this paper uses RNN to realize the pattern recognition module instead of ANN, CNN, DNN, and GNN.

RNN has many parameters which face the problem of weight assignment. The common methods for weight assignment include traditional exact methods, mathematical heuristic methods, and intelligent bio-inspired methods. The traditional exact methods cannot adapt to the large-scale scenario. The mathematical heuristic methods have good computation efficiency under the large-scale scenario while the obtained solution is usually not optimal. Thus, this paper uses genetic algorithm (GA) [21] to do weight assignment for the involved parameters.

According to the above statements, this paper further investigates data pre-processing and pattern recognition modules of electronic nose, depending on AI technique. Regarding the data pre-processing module, LLE is used to reduce data dimensionality. Regarding the pattern recognition module, RNN is used to realize its function. In terms of the problem that RNN has many parameters, GA is employed to do weight assignment for those parameters. To sum up, the major contributions of this paper are as follows: (1) LLE is used to make dimensionality reduction to avoid information redundancy; (2) RNN

is used to do pattern recognition, where GA is employed to adjust weight; and (3) based on the real sensor drift dataset of electronic nose, three metrics, i.e., accuracy rate, response time, and stability, are verified.

The remaining of this paper is organized as follows. Section 2 reviews the related research work from two perspectives. Section 3 introduces dimensionality reduction based on LLE. Section 4 presents RNN-based pattern recognition. Section 5 reports the experimental results. Section 6 concludes this paper.

## 2. Related Work

There have been a lot of studies on electronic nose, including the related applications and the inherent technologies.

### 2.1. The Related Applications

As the mentioned in Introduction, electronic nose shows the wide applications in all fields. For example, in [22], a model transfer learning framework with back-propagation neural network for win and Chinese liquor detection by electronic nose was proposed. In [23], a deep feature mining method of electronic nose sensor data for identifying beer olfactory information was proposed. References [22,23] indicated that electronic nose can be used to detect ethyl alcohol. In addition, as the classical applications, electronic nose can also be applied to do gas recognition. For example, in [24], a drift-compensating novel deep belief classification network was devised to improve gas recognition of electronic nose. In [25], a minimum distance inlier probability feature selection method was presented to improve gas classification for the electronic nose system. In [26], an efficient electronic nose system for odor analysis and assessment was designed, where the fault detection and alarming design could generate a high-reliability performance by constantly monitoring the working status. In addition, electronic nose has good detection performance on formalin. For example, in [27], formalin fresh noodles with electronic nose based on kernel principal component analysis was introduced. In [28], formalin on fresh tilapia via electronic nose and assessment of toxicity levels with reference to average adult Filipino weight was proposed.

Furthermore, electronic nose also has more advanced applications. For example, in [29], the authors made good optimization of extracted features for an explosive-detecting electronic nose by using GA. In [30], tofu shelf life was monitored by using electronic nose based on curve fitting method. In [31], the authors presented an overview of the most important contributions dealing with the quality control in microbial fermentation process by using electronic nose. In [32], an electronic nose-based assistive diagnostic prototype for lung cancer detection with conformal prediction was proposed. In [33], citrus tristeza virus in mandarin orange was detected by using a custom-developed electronic nose system. In [34], feature extraction of citrus juice during storage for electronic nose based on cellular neural network was developed. In [35], a novel quality evaluation method for magnolia bark was proposed by using electronic nose and colorimeter data with the multiple statistical algorithms. In [36], the authors made the comprehensive research on principles and recent advances in electronic nose for quality inspection of agricultural and food products. In [37], an optimized deep CNN for dendrobium classification based on electronic nose was proposed. In [38], on-line assessment of oil quality during deep frying was addressed by using an electronic nose and proton transfer reaction mass spectrometry. In [39], a novel method for rapid quantitative evaluating formaldehyde in squid based on electronic nose was devised. In [40], quality grades of organic green teas was classified and evaluated by using electronic nose based on machine learning algorithms. In [41], the authors made early detection of fish degradation by electronic nose.

### 2.2. The Inherent Technologies

Even though the above-reviewed applications show nice performance and obtain general acceptance, they usually neglect the inherent technique improvement of electronic nose. To this end,

some solutions regarding this have been proposed. For example, in [42], a novel technique to solve shortages of low-concentration samples of electronic nose based on global and local features fusion was presented. In [43], a natural neural learning model inspired electronic nose system was devised. To be specific, a natural on-line training with only one sample, to extract both eigen-weights and eigen-bias, was built to elaborate a natural identifier neural model in a real work environment. The proposed model efficiently could reduce the maximum extent of traditional neural models complexities, namely generic work-laboratory, dimensional data learning, model adaptability complication, time-consuming, heavy experiment materials, and chemical products. In [44], the authors proposed a sensor drift correction method based on discriminative subspace projection to deal with the sensor drift problem. In [45], the authors employed manifold learning algorithms to improve the classification performance of electronic nose. In [46], multi-sensor electronic nose based on conformal sensor chamber was designed. In [47], the adaptive subspace learning was used to make drift compensation for electronic nose. In [48], drift compensation for electronic nose by multiple classifiers system with GA optimized feature subset was solved. In [49], fuzzy c-means clustering based novel threshold criteria for outlier detection in electronic nose was proposed. In [50], the joint distribution adaptation for drift correction in electronic nose type sensor array was presented. In [51], online drift compensation by the adaptive active learning on mixed kernel for electronic nose was proposed, which depended on an assumption that the calibration samples were gained online with uncertain amount. It redesigned a hybrid sample-evaluation kernel assessing samples comprehensively by introducing a ranking method to normalize the outputs of kernel. In [52], ANN was used to process electronic nose data. In [53], the authors discussed the training technique of electronic nose by using the labeled and unlabeled samples based on multi-kernel support vector machine (SVM). In [54], the rapid detection approach for enhancing the electronic nose system's performance was verified by using different deep learning models and SVMs, where three deep learning architecture implementations types were used for the classification tasks. Among them, the first deep learning model was implemented employing machine learning framework; the second architecture implementation type was to perform meta-learning, adjusting the connections between different computing cells by differentiable search to obtain the best graph configuration while training; the final model corresponded to a simple multilayer perceptron with the fully connected layers.

　　Without a doubt, although these technologies improve the performance of electronic nose system, they still have a great optimization space, such as accuracy rate, response time, and stability. Furthermore, different from the current studies, this paper optimizes the electronic nose system from the perspective of data pre-processing and pattern recognition. The mentioned two aspects motivate this paper.

## 3. LLE-Based Dimensionality Reduction

　　In terms of electronic nose system, the dimensionality reduction of data plays an important role to improve computation efficiency and guarantee computation accuracy. The dimensionality reduction of data is defined as follows.

**Definition 1.** *Given N feature vectors, i.e., $\{x_1, x_2, \cdots, x_N\}$, here $\forall x_i$ ($i \in [1, N]$) is d-dimensional space and $x_i \in R^d$, and the feature vectors after dimensionality reduction are $\{y_1, y_2, \cdots, y_N\}$ with m-dimensional space, satisfying $y_i \in R^m$ and $m \ll d$.*

　　Compared with other dimensionality reduction methods, LLE has faster computation speed and more accurate computation result. Therefore, this paper employs LLE for dimensionality reduction, which usually includes three parts: graph construction, weight determination, and data mapping.

## 3.1. Graph Construction

This paper adopts K-nearest neighbor (KNN) algorithm [55] to construct the graph with respect to all feature vectors, that is to say, for $\forall x_i$, its $K$ nearest neighbors (i.e., data points) need to be found. The core idea is described as three steps. First, for $\forall x_i$, the distance between it and each $x_j$ ($i \neq j$) is computed, i.e., $N - 1$ distance values are obtained. Then, these distance values are arranged in the descending order. Finally, the first $K$ data points are regarded as the nearest neighbors of $x_i$.

However, the determination of $K$ is key but difficult. To be specific, if $K$ is set as relatively small, it means that the whole model becomes complex and easily causes overfitting. On the contrary, if $K$ is set as relatively large, it means that the whole modes becomes simple and dimensionality reduction cannot reach the satisfactory effect. With such consideration, this paper determines $K$ according to the distribution of sample data points. Let $d_{i,j}$ denote the distance between $x_i$ and $x_j$, and $N - 1$ distances can be obtained as follows: $d_{i,1}, d_{i,2}, \cdots, d_{i,N-1}$. Regarding the $N - 1$ distances, the corresponding mean and variance can be defined as follows.

$$\mu = \frac{1}{N-1} \sum_{j=1}^{N-1} d_{i,j} \tag{1}$$

$$\sigma^2 = \frac{1}{N-1} \sum_{j=1}^{N-1} (d_{i,j} - \mu_i)^2 \tag{2}$$

Suppose that the distance between sample data points and the current using data points follows the Gaussian distribution, and the improved $K$ is defined as follows.

$$K = f(|\mu - \xi * \sigma|) \tag{3}$$

Among them, $f(x)$ is the number of feature vectors where the distance is smaller than $x$, and $\xi$ is a parameter. In particular, when $\xi = 3$, the coverage rate in the interval $[\mu - 3\sigma, \mu + 3\sigma]$ can reach the maximum value, i.e., 99.73%.

## 3.2. Weight Determination

For all $x_i$ and $K_i$, it is required to build a matrix with respect to the local weight values while guaranteeing the corresponding construction error reaches the minimal value. Let $W$ and $\epsilon(W)$ denote such matrix and such construction error, and $\epsilon(W)$ is defined as follows.

$$\epsilon(W) = \sum_{i=1}^{N} \left\| x_i - \sum_{j=1}^{K} w_{i,j} x_j \right\|^2 \tag{4}$$

where $x_j$ is a neighbor of $x_i$ and $w_{i,j}$ is the weight between $x_i$ and $x_j$. In particular, $\sum_{j=1}^{K} w_{i,j} = 1$ is satisfied. Furthermore, Equation (4) is converted as follows.

$$\epsilon(W) = \sum_{i=1}^{N} \left\| \sum_{j=1}^{K} w_{i,j} (x_i - x_j) \right\|^2 = \sum_{i=1}^{N} W_i^T (x_i - x_j)^T (x_i - x_j) W_i \tag{5}$$

Put $W_i = (w_{i,1}, w_{i,2}, \cdots, w_{i,d})^T$ and $Z_i = (x_i - x_j)^T (x_i - x_j)$ into Equation (5), and Equation (5) is simplified as follows.

$$\epsilon(W) = \sum_{i=1}^{N} W_i^T Z_i W_i \tag{6}$$

According to the Lagrange multiplication, a new equation is obtained as follows.

$$L(W) = \sum_{i=1}^{N} W_i^T Z_i W_i + \lambda(W_i^T 1_d - 1) \tag{7}$$

where, $1_d$ is a $d$-dimensional vector with all values for 1. The derivation operation is performed in terms of $W$, and then the derivative result is set as 0. The following equation is obtained.

$$2Z_i W_i + \lambda 1_d = 0 \tag{8}$$

Put $W_i^T 1_d = 1$ into Equation (8), and $W_i$ can be obtained as follows.

$$W_i = \frac{Z_i^{-1} 1_d}{1_d^T Z_i^{-1} 1_d} \tag{9}$$

According to the above formula manipulation, the weight values can be obtained by minimizing the construction error, which has an important property: translation, rotation, and zoom operations have no influence on the weight determination.

*3.3. Data Mapping*

To guarantee the topology structure consistency of data points between high-dimensional space and low-dimensional space as much as possible, it is required to build a cost function while satisfying the minimal cost function value. Let $\Phi(Y)$ denote such cost function, $y_i$ denote the output result of $x_i$, and $\Phi(Y)$ be defined as follows.

$$\Phi(Y) = \sum_{i=1}^{N} \left\| y_i - \sum_{j=1}^{K} w_{i,j} y_j \right\|^2 \tag{10}$$

where $y_j$ is a neighbor of $y_i$. In particular, the following two constraint conditions are satisfied.

$$\sum_{i=1}^{N} y_i = 0, \frac{1}{N} \sum_{i=1}^{N} y_i y_i^T = I \tag{11}$$

where $I$ is the unit matrix. Furthermore, $\Phi(Y)$ can be simplified as follows.

$$\Phi(Y) = tr\left(Y_T (I - W)^T (I - W) Y\right) \tag{12}$$

Similarly, according to the Lagrange multiplication, a new equation is obtained as follows.

$$L(Y) = tr(Y^T M Y) + \lambda(Y^T Y - dI) \tag{13}$$

where $M$ is symmetric matrix for $N \times N$. The derivation operation is performed in terms of $Y$, and then the derivative result is set as 0. The following equation is obtained.

$$MY = \lambda Y \tag{14}$$

On this basis, for the smallest $m$, feature values are computed, and their corresponding feature vectors $y_1, y_2, \cdots, y_N$ are the final solution.

## 4. RNN-Based Pattern Recognition

In terms of the electronic nose system, the pattern recognition module is the most important part and has a direct influence on accuracy rate and stability. This paper uses RNN to realize the

pattern recognition module, including RNN introduction and GA-based weight assignment for the involved parameters.

*4.1. LSTM-Based RNN*

As is known, there exists the vanishing gradient problem for the traditional RNN, and thus this paper employs long short-term memory (LSTM) [56] for RNN to address the vanishing gradient problem. Each LSTM unit has hidden state $h_t$, memory unit $c_t$, and three gates (i.e., input gate $i_t$, forget gate $f_t$, and output gate $o_t$). Besides, each gate is activated by the sigmoid function, generating the corresponding values between 0 and 1 as follows.

$$i_t = s(W_i v_t + U_i h_{t-1} + b_i) \tag{15}$$

$$f_t = s(W_f v_t + U_f h_{t-1} + b_f) \tag{16}$$

$$o_t = s(W_o v_t + U_o h_{t-1} + b_o) \tag{17}$$

where $s(\cdot)$ denotes the sigmoid function; $W$ and $U$ are two kinds of weight matrixes; and $b$ is the offset. In particular, the time step length $t$, the collected attribute of electronic nose $v_t$, the previous hidden state $h_{t-1}$, and the previous memory unit $c_{t-1}$ are considered as the inputs of LSTM as follows.

$$g_t = \tan h(W_g v_t + U_g h_{t-1} + b_g) \tag{18}$$

$$c_t = f_t \odot c_{t-1} + i_t \odot g_t \tag{19}$$

$$h_t = o_t \odot \tan h_{c_t} \tag{20}$$

In fact, LSTM usually has two types, denoted by $\overrightarrow{LSTM}$ and $\overleftarrow{LSTM}$. $\overrightarrow{LSTM}$ denotes the LSTM with the forward calculation and $\overleftarrow{LSTM}$ denotes the LSTM with the reverse calculation. Then, two corresponding hidden states are defined as follows.

$$\overrightarrow{h_t} = \overrightarrow{LSTM}(v_t, \overrightarrow{h_{t-1}}) \tag{21}$$

$$\overleftarrow{h_t} = \overleftarrow{LSTM}(v_t, \overleftarrow{h_{t-1}}) \tag{22}$$

*4.2. GA-Based Weight Assignment*

As can be seen from Equations (15)–(18), there are many parameters waiting for weight assignment in RNN. Considering that GA has the global optimization performance for these weight parameters, this paper uses GA to do the weight assignment [21]. Regarding this, the objective function can be written as follows.

$$MinimizeF = \alpha(W_i + W_f + W_o + W_g) + \beta(U_i + U_f + U_o + U_g) \tag{23}$$

The next is to solve Equation (23) via GA. In particular, GA is used to obtain the relatively optimal path and these nodes in the determinate path build a network topology with some designated weight values. Meanwhile, these weight values are considered as a weight assignment solution. GA usually consists of selection operator, crossover operator, variation operator, and fitness function. Different from the traditional GA, this paper presents an adaptive GA to do the automatic selection operator. To be specific, this paper designs an online adjustment method on selection pressure to make sure the tradeoff between fast convergence and population diversity. For the arbitrary individual $i$, let $p_i$ denote its selection probability. When the nonlinear relationship is considered, $p_i$ is defined as follows.

$$p_i = \gamma(1 - \gamma)^{r_i} \tag{24}$$

where $\gamma$ is the coefficient of pressure control and $r_i$ is the rank of individual $i$. Furthermore, let $M$ denote the initial population size, and the best individual's section probability and the worst individual's section probability are defined as follows.

$$p_{best} = \gamma, p_{worst} = \gamma(1-\gamma)^{M-1} \tag{25}$$

It is obvious that the determination of $\gamma$ is very important. Given that, this paper uses the standard deviation with respect to all individual fitness values to determine $\gamma$. First, the standard deviation $sd$ is defined as follows.

$$sd = \sqrt{\frac{1}{M}\sum_{i=1}^{M}(fit_i - fit_{ave})^2} \tag{26}$$

where $fit_i$ is the fitness of individual $i$ and $f_{ave}$ is the average value. Then, for the $T$th iteration, its corresponding $\gamma$ is defined as follows.

$$\gamma_T = \begin{cases} \gamma_{T-1} - 0.05, sd < thr_1 \\ \gamma_{T-1}, thr_1 \leq sd \leq thr_2 \\ \gamma_{T-1} + 0.05, sd > thr_2 \end{cases} \tag{27}$$

where $thr_1$ and $thr_2$ are two parameters. Especially when $sd > thr_2$, it needs to adjust $\gamma$ to be large so that the selection probability of individual can be increased.

## 5. Results

### 5.1. Dataset Collection

The sensor drift dataset comes from [57]. In total, 1604 samples were collected by using the multiple E-nose devices with the same model. Besides, the dataset consists of three batches, i.e., batch master collected five years earlier than the batches slave 1 and slave 2. Meanwhile, there are six kinds of gases to be detected: ammonia, benzene, carbon monoxide, formaldehyde, nitrogen dioxide, and toluene. The detailed information of the dataset is shown in Table 1.

**Table 1.** The sensor drift dataset.

| Batch | Ammonia | Benzene | Carbon Monoxide | Formaldehyde | Nitrogen Dioxide | Toluene | Total |
|---|---|---|---|---|---|---|---|
| Master | 60 | 72 | 58 | 126 | 38 | 66 | 420 |
| Slave 1 | 81 | 108 | 98 | 108 | 107 | 106 | 608 |
| Slave 2 | 84 | 87 | 95 | 108 | 108 | 94 | 576 |

### 5.2. Experiment Method

The experiments included two parts. The first part was the performance analysis of RNN, testing recall ratio, precision ration, and F1 value. Then, the second part was the comparison analysis, testing accuracy rate, response time, and stability. Meanwhile, three benchmarks were selected from the latest research achievements [43,51,54]. Ref. [43] presented a natural neural learning model inspired electronic nose system, called NNL; ref. [51] proposed an online drift compensation by the adaptive active learning on mixed kernel for electronic nose, called AAL; and ref. [54] used different deep learning models and SVMs to enhance the electronic nose system's performance, called DLS. In addition, the involved parameters were set as follows: $M = 100$, $\alpha = 0.45$, $\beta = 0.55$, and the number of simulation times was 10. In particular, the feature extraction was performed for each sensor resulting in a six-dimensional feature vector for each sample.

### 5.3. RNN Performance Analysis

The average recall ratios regarding six kinds of gases are shown in Table 2. The average precision ratios regarding six kinds of gases are shown in Table 3. The average F1 values regarding six kinds of gases are shown in Table 4.

**Table 2.** The average recall ratios for the six kinds of gases (%).

| Batch | Ammonia | Benzene | Carbon Monoxide | Formaldehyde | Nitrogen Dioxide | Toluene |
|---|---|---|---|---|---|---|
| Master | 98.26 | 97.38 | 98.29 | 97.68 | 98.16 | 98.22 |
| Slave 1 | 95.87 | 96.19 | 97.46 | 96.89 | 97.37 | 96.95 |
| Slave 2 | 96.76 | 97.13 | 96.55 | 97.61 | 97.08 | 98.33 |

**Table 3.** The average precision ratios for the six kinds of gases (%).

| Batch | Ammonia | Benzene | Carbon Monoxide | Formaldehyde | Nitrogen Dioxide | Toluene |
|---|---|---|---|---|---|---|
| Master | 96.97 | 97.83 | 96.09 | 97.86 | 97.66 | 96.58 |
| Slave 1 | 96.75 | 95.28 | 98.12 | 99.03 | 97.64 | 98.59 |
| Slave 2 | 97.38 | 97.19 | 96.28 | 96.79 | 98.32 | 98.76 |

**Table 4.** The average F1 values for the six kinds of gases (%).

| Batch | Ammonia | Benzene | Carbon Monoxide | Formaldehyde | Nitrogen Dioxide | Toluene |
|---|---|---|---|---|---|---|
| Master | 97.61 | 97.60 | 97.18 | 97.77 | 97.91 | 97.39 |
| Slave 1 | 96.31 | 95.73 | 97.79 | 97.95 | 97.50 | 97.76 |
| Slave 2 | 97.07 | 97.16 | 96.41 | 97.20 | 97.70 | 98.54 |

In Tables 1–3, we observe that the used LSTM-based RNN shows good recall ratio, precision ratio, and F1 value, as all related values could reach 95%. It also indicates that using RNN to realize pattern recognition module is feasible.

### 5.4. Comparison Analysis

The average accuracy rates of the proposed method, NNL, AAL, and DLS, under different groups of experiments are shown in Figure 1. The average response times of the proposed method, NNL, AAL, and DLS, under different groups of experiments, are shown in Figure 2.

We found that our method presents the highest accuracy rate and the lowest response time. This indicates that LLE-based dimensionality reduction and RNN-based pattern recognition can greatly improve the electronic nose system, while obtaining the accurate detection results with low response time.

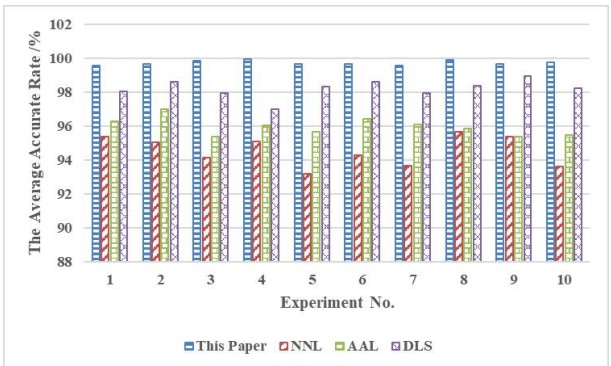

**Figure 1.** The average accuracy rates of the four methods under different experiments.

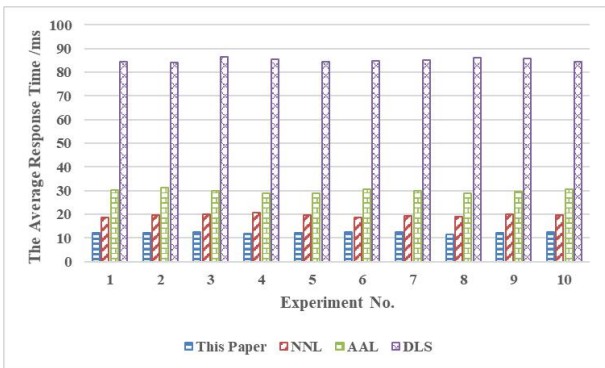

**Figure 2.** The average response times of the four methods under different experiments.

Furthermore, the standard deviation was used to measure the stability. For two metrics, i.e., accuracy rate and response time, the corresponding standard deviation values in terms of 10 different experiments were computed by Equation (26). A smaller standard deviation value means a higher stability. The experimental results are shown in Figure 3.

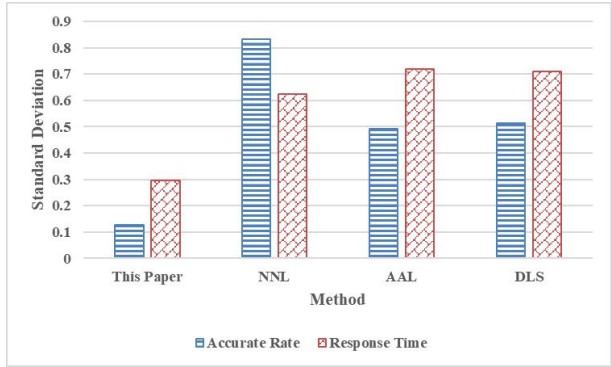

**Figure 3.** The standard deviation values for all experiments.

It is obvious that the method proposed in this paper has the smallest standard deviation values in terms of both accuracy rate and response time, which further indicates that it has the highest stability.

*5.5. Discussion*

Since the experimental results are based on the phase of virtual simulation rather than the implemented product, the validation carried out has some threats. For intrinsic reasons, there are three aspects. First, the weight assignment based on GA has different influences in terms of different datasets, i.e. the fixed weight assignment does not mean that the proposed optimization method in this paper can obtain the optimal solution for all datasets. Second, the building of RNN structure can be dynamic and it may be unstable during the process of data training. Third, the written code is probably unstable and even redundancy exists, which has an important influence on the computational efficiency. For the extrinsic reasons, they include two aspects. On the one hand, the adopted datasets lack diversity, and the current experimental results can only demonstrate that the proposed optimization method is efficient within a certain range but cannot guarantee that it is forever efficient, after all it is not a mass-produced product. On the other hand, different coding styles also have a considerable influence on the experimental results. For example, RNN is coded in C++ language and the electronic nose system is implemented in C language.

**6. Conclusions**

The whole electronic nose system usually includes gas sensor, data pre-processing, and pattern recognition modules. Currently, most studies pay attention to the applications of electronic nose

irrespective of the inherent technique improvement of electronic nose. Although there are some proposals to optimize the inherent technique, they usually focus on modifying the gas sensor module and barely pay attention to the improvement of data pre-processing module and pattern recognition module, which are addressed in this paper. First, LLE is employed for dimensionality reduction, including graph construction, weight determination, and data mapping. Then, RNN is used for realizing the pattern recognition module. In particular, LSTM is adopted to improve RNN and GA is leveraged to do the weight assignment for the involved parameters. The experiments are implemented based on the real sensor drift dataset, which include two parts: RNN performance analysis and comparison analysis. The first part tested recall ratio, precision ration, and F1 value, which can reach 95%. The second part tested accuracy rate, response time, and stability. It was found that this method has the best optimization performance on the electronic nose system.

**Author Contributions:** Conceptualization, Y.Z.; methodology, J.L.; software, Y.Z. and J.L.; formal analysis, J.L.; investigation, Y.Z.; resources, Y.Z. and J.L.; writing—original draft preparation, Y.Z.; writing—review and editing, Y.Z. and J.L.; and funding acquisition, Y.Z. All authors have read and agreed to the published version of the manuscript.

**Funding:** This work was supported by the project of Education Department of Jilin Province in China under grant JJKH20190833KJ.

**Conflicts of Interest:** The authors declare no conflict of interest.

## Abbreviations

The following abbreviations are used in this manuscript:

| | |
|---|---|
| NN | Neural Network |
| ANN | Artificial NN |
| CNN | Artificial NN |
| DNN | Deep NN |
| GNN | Graph NN |
| RNN | Recurrent Neural Network |
| LLE | Locally Linear Embedding |
| AI | Artificial Intelligence |
| PCA | Principal Component Analysis |
| LDA | Linear Discriminant Analysis |
| LE | Laplacian Eigenmaps |
| t-SNE | t-Stochastic Neighbor Embedding |
| GA | Genetic Algorithm |
| SVM | Support Vector Machine |
| KNN | K-Nearest Neighbor |
| LSTM | Long Short-Term Memory |
| NNL | Natural Neural Learning |
| AAL | Adaptive Active Learning |
| DLS | Deep Learning models and SVMs |

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
