# Peer review of "Using Recurrent Neural Network to Optimize Electronic Nose System with Dimensionality Reduction"

_electronics, doi:10.3390/electronics9122205_

Round 1

Reviewer 1 Report

The paper presents an interesting approach to optimize the electronic nose system by using Recurrent Neural Network for the pattern recognition and Locally Linear Embedding for dimensionality reduction of high-dimensional data. 

Section 1 gives good introduction to the problem and lays out and compares the different methods used for data pre-processing as well as pattern recognition. Section 2 describes some related works where an electronic nose system has been employed and the methods used in those publications.

Section 3 & 4 describe the LLE based dimensionality reduction and the RNN based pattern recognition. The detailed description of these methods is useful however these sections are based on a lot of already known material. As such these sections could be reduced. 

Section 5 shows encouraging results from this method, especially when compared to other approaches.

The only concern I have is that the paper describes the RNN and LLE methods in lengthy detail. These are known methods, and as such the lengthy description doesn’t add new information.

Author Response

Response: Thanks for your comment. Although RNN and LLE are well-known methods, they are the key components of this paper. Thus, we think that it is necessary to provide the information of RNN and LLE for readers. In particular, different from the others methods for the optimization of electronic nose system, RNN and LLE are the novel applications.  

Reviewer 2 Report

The paper deals with a very interesting, from the scientific point of view, problem concerning research in the field of electronic nose. In recent years, issues related to the development or improvement of the electronic nose have found a permanent place in the literature.

The paper proposes optimization of the electronic nose system from the point of view of data pre-processing and pattern recognition. Main focus was put on improving the technique of electronic nose operation. This is a valuable aspect of the presented article. The local linear deposition (LLE) method was used to reduce multidimensionality, while the recursive neural sequence (RNN) was used to recognize patterns and guarantee accuracy and stability.

All experiments are very well documented, experimental results confirm efficiency of usage of these networks.

A solid bibliography review was conducted in the paper. The authors point  to the most modern solutions in a described field.  A couple of bibliography items, quoted in the article, were issued in the year 2020.

Author Response

Response: We would like to thank you for appreciating this paper.

Reviewer 3 Report

It is a good research article, interesting, relevant and well presented. It still has four aspects that I believe need to be improved:
- The introduction can be improved to better explain the motivation of this work, the objectives and above all applications and generalizations of the research carried out.
- The scientific method applied in the resolution is not well explained. I would restructure the initial part presenting clearly what research questions are posed to be solved and how they are solved.
- In the validation, results are presented against NNL, AAL, DLS... but in section 2 these solutions are not detailed in depth as they should be. I propose to add an SLR (systematic literature review) where similar solutions are presented and a more detailed explanation of them.
- I propose to add a section discussing the threats to the validation carried out (intrinsic, extrinsic...) as this validation is subject to many limitations and threats to the validity of the data presented.

Author Response

  1. The introduction can be improved to better explain the motivation of this work, the objectives and above all applications and generalizations of the research carried out.

Response: Thanks for this comment.

At first, the concept, research field and research limitations of electronic nose are introduced in paragraph 1. To be specific, (1) electronic nose is introduced from lines 19 to 26); (2) the research field of electronic nose is introduced from lines 27 to 30; (3) the limitations of electronic nose are introduced as “To the best of our knowledge, there are also some proposals to optimize the inherent technique, but they usually focus on modifying the module of gas sensor and barely pay attention to the improvement on data pre-processing and pattern recognition.” (see lines 30-33).

Secondly, paragraph 2 states the importance of further studying data pre-processing and pattern recognition modules of electronic nose, depending on AI technique.

Thirdly, paragraph 3 introduces that the data pre-processing module’s main responsibility is to do dimensionality reduction for high-dimensional data features extracted from gas sensor array. Furthermore, some methods for dimensionality reduction are introduced and finally LLE is demonstrated with the highest rationality.

Fourthly, paragraph 4 conducts that RNN is a nice method to realize the pattern recognition module in electronic nose system.

Fifthly, in terms of the problem which RNN has many parameters for weight assignment, paragraph 5 states that GA is a nice method to do weight assignment for these involved parameters.

Finally, paragraph 6 summarize the contributions of this paper, including three aspects. (see lines 81-85).

To sum up, the introduction shows the strong logic.

  1. The scientific method applied in the resolution is not well explained. I would restructure the initial part presenting clearly what research questions are posed to be solved and how they are solved.

Response: In lines 43-45, we have stated that the research questions are to further study data pre-processing and pattern recognition modules of electronic nose, depending on AI technique. Furthermore, in lines 81-82, we further declared the research questions as follows: “this paper optimizes electronic nose system from the perspective of data pre-processing and pattern recognition.” Regarding the scientific method, in terms of data pre-processing, LLE is used to make dimensionality reduction; in terms of pattern recognition, RNN is employed. These have been declared in lines 83-85. For LLE and RNN, they are detailedly introduced in Section 3 and Section 4 respectively.

  1. Inthe validation, results are presented against NNL, AAL, DLS... but in section 2 these solutions are not detailed in depth as they should be. I propose to add an SLR (systematic literature review) where similar solutions are presented and a more detailed explanation of them.

Response: Thanks for this comment. In Section 2, we have given more illustrations on NNL [43], AAL [51] and DLS [54], as follows:

In [43], a natural neural learning model inspired electronic nose system was devised. To be specific, a natural on-line training with only one sample, to extract both eigen-weights and eigen-bias was built herein to elaborate a natural identifier neural model in a real work environment. The proposed model efficiently could reduce the maximum extent of traditional neural models complexities, namely generic work-laboratory, dimensional data learning, model adaptability complication, time-consuming, heavy experiment materials and chemical products.

In [51], online drift compensation by adaptive active learning on mixed kernel for electronic nose was proposed, which depended on an assumption that the calibration samples were gained online with uncertain amount by category and the recognition-learner updating performs based on few calibration samples allowed to query their categories. It redesigned a hybrid sample-evaluation kernel assessing samples comprehensively by introducing a ranking method to normalize the outputs of kernel.

In [54], the rapid detection approach for enhancing the electronic nose system's performance was verified by using different deep learning models and SVMs, where three deep learning architecture implementations types were used for the classification tasks. Among them, the first deep learning model is implemented employing machine learning framework; the second architecture implementation type was to perform meta-learning, adjusting the connections between different computing cells by differentiable search to obtain the best graph configuration while training; the final model corresponded to a simple multilayer perceptron with only fully connected layers.

  1. I propose to add a section discussing the threats to the validation carried out (intrinsic, extrinsic...) as this validation is subject to many limitations and threats to the validity of the data presented.

Response: In the revised version, we have added a section (Section 5.5, page 10) to discuss the threats to the validation carried out, as follows:

Since the experimental results are based on the phase of virtual simulation rather than the implemented product, the validation carried out has some threats. For the intrinsic reasons, there are three aspects. At first, the weight assignment based on GA has different influences in terms of different datasets, that is to say, the fixed weight assignment does not means that the proposed optimization method in this paper can obtain the optimal solution for all datasets. Secondly, the building of RNN structure can be dynamic and it may be unstable during the process of data training. Thirdly, the written code is probably unstable and even exits the redundancy, which have the important influence on computation efficiency. For the extrinsic reasons, they include two aspects. On on hand, the adopted datasets lack of diversity, and the current experiment results only can demonstrate that the proposed optimization method is efficient within a certain range but cannot guarantee that it is forever efficient after all it is not a mass-produced product. On the other hand, different coding styles also have the considerably influence on the experimental results. For example, RNN is coded in C++ language and electronic nose system is implemented in C language.

Round 2

Reviewer 3 Report

Unfortunately, of the 4 points of improvement I raised only one of them (the last one) has been considered in this second version, and only partially. A more detailed explanation of the contents do not change my initial opinion after reading the same contents, I think that where I indicated that there is opportunity for improvement there is still opportunity, and the authors have only been devoted to shell line by line where they comment on points that I have indicated as improvable, without adding new information that actually results in better content for the article. Therefore, and simply for consistency with my own review, I cannot alter my initial opinion.

Author Response

We would like to thanks these comments.

  1. For the first comment, this introduction has shows the strong organization logic, where the motivation of this work, the objectives and research contributions have been explained well. Therefore, we think that there is no need to make the further improvement.

  1. For the second comment, we have re-organized the second paragraph from bottom of section 1, as follows:

According to above statements, this paper further investigates data pre-processing and pattern recognition modules of electronic nose, depending on AI technique. Regarding data pre-processing module, LLE is used to reduce data dimensionality. Regarding pattern recognition module, RNN is used to realize its function. In terms of the problem which RNN has many parameters, GA is employed to do weight assignment for these involved parameters. To sum up, the major contributions of this paper are concluded as follows. (i) LLE is used to make dimensionality reduction to avoid information redundancy. (ii) RNN is used to do pattern recognition, where GA is employed to adjust weight; (iii) Based on the real sensor drift dataset of electronic nose, three metrics, i.e., accurate rate, response time, and stability are verified.

  1. For the third comment, these three baselines, i.e., [43], [51] and [54] have beenpresented in the more detailed way in the current version.

  1. As a application innovation on electronic nose, LLE and RNN have presented the nice application implementation. In addition, we think organization logic, presentation, writing and structure deployment are very acceptable, which have also been approved by the other two reviewers. Therefore, we sincerely hope you could accept this paper.